# Concrete Shrinkage Analysis with Quicklime, Microfibers, and SRA Admixtures

**DOI:** 10.3390/ma16052061

**Published:** 2023-03-02

**Authors:** Daumantas Židanavičius, Mindaugas Augonis, Nerijus Adamukaitis, Ignacio Villalon Fornes

**Affiliations:** Faculty of Civil Engineering and Architecture, Kaunas University of Technology, Studentu Str. 48, LT-51367 Kaunas, Lithuania

**Keywords:** concrete shrinkage, long-term monitoring, quicklime, humidity, shrinkage prediction, B4 model, Le-Chatelier

## Abstract

This research explores the effect of various humidity conditions and the efficiency of shrinkage-reducing admixtures on the free shrinkage strain of ordinary Portland cement (OPC) concrete and its mechanical properties. An OPC concrete C30/37 mixture was replenished with 5% of quicklime and 2% of organic-compound-based liquid shrinkage-reducing agent (SRA). The investigation revealed that a combination of quicklime and SRA led to the highest reduction in concrete shrinkage strain. Polypropylene microfiber addition was not so effective in reducing the concrete shrinkage as the previous two additives did. The prediction of concrete shrinkage without quicklime additive was performed according to EC2 and B4 model methods, and the obtained results were compared with the experimental ones. The B4 model evaluates the parameters more than the EC2 model does and, therefore, was modified to calculate the concrete shrinkage for the case of variable humidity and to evaluate the effect of quicklime additive. The experimental shrinkage curve that best coincides with the theoretical one was that obtained by the modified B4 model.

## 1. Introduction

Concrete shrinkage can be free or restrained. In the case of free shrinkage, the concrete does not suffer any cracking or additional stresses. On the other hand, restrained shrinkage is typical of reinforced concrete structures, as the reinforcement restrains the free concrete shrinkage, which causes tensile stresses appearing in concrete. Concrete cracking takes place if the tensile stresses exceed its tensile strength [1]; therefore, a higher amount of reinforcement causes higher tensile stresses in concrete. A slightly different concrete shrinkage behavior occurs in composite steel and concrete structures. In this case, the bond stresses (not only the normal ones) are caused by shrinkage [2]. Moreover, it is widely known that the shrinkage of concrete is a property depending on several factors, such as the type and amount of binder, the amount of mixing water, and the humidity of the environment. While the type of binder and the mixing water content are factors of the concrete mixture that does not change over time, the environment humidity does, and considerably differs for indoor and outdoor conditions. The lower humidity causes higher shrinkage. Many models have been created to predict the concrete shrinkage for the case of constant humidity [3]. However, this assumption does not apply for partially closed composite steel-concrete structures, as the humidity escapes through their open ends. Therefore, the humidity in a partially closed section will decrease with time [4]. However, the vast majority of the experimental shrinkage studies and theoretical models consider the humidity as a time-invariable parameter. Bazant [5] found that shrinkage at cyclic humidity does not exhibit any systematic difference from the depicted shrinkage at average humidity. However, for the case of a permanent decrease in humidity, the concrete shrinkage may not be comparable to that at average humidity. It is possible that, at the early stages of curing, the effect of humidity on concrete shrinkage may be more significant, as, at this stage, the degree of hydration is higher. The long-term shrinkage effect occurs 24 h after concrete mixture blending and is attributed to water exchange and evaporation [6,7,8,9]. Therefore, in the current study, the model B4 was modified for theoretical assessment of the influence of humidity on concrete shrinkage. The experiments were performed and comparative calculations were made by the EC2 method to verify this modified model. Due to practical analysis, the most widely used shrinkage-reducing admixtures (SRAs) were used in the experiments. There are various types of admixtures, such as fibers, microfibers, absorbent polymers, and supplementary cementitious materials (SCMs)—fly ash [9,10,11,12,13,14]. Hence, it was decided to use several types of admixtures to evaluate their impact on the concrete shrinkage.

The effect of different types of fibers on the concrete shrinkage can be significant and depends on the concrete mixture [10,11,12]. Ullah et al. [13] found that a certain amount of polypropylene fibers produced a more positive effect on autogenous shrinkage than that produced by steel or glass fibers. As the polypropylene fiber is lighter and more economic, it was selected for the current research [14,15,16,17,18,19].

Fly ash (FA) is widely used in hydraulic mass concrete structures to reduce the temperature rise and the resulting thermal shrinkage. Overall, FA is a relatively expensive material and often used with low-heat Portland cement (LHP). Such concrete mixtures are utilized in concrete-resulting face slabs or concrete-faced rockfill dams [20]. In addition, fly ash is more widely used in self-compacting concrete and rarely in structural concrete. Moreover, the fly ash reduces the expansion, which is not an advantage for partially closed steel concrete structures. It is well known that some materials such as admixtures can cause expansion of concrete. Quicklime is one of these that has this property. The expansion at the early stage of binding can partially or even fully compensate for the shrinkage of concrete [8,21,22,23,24,25,26,27,28]. Hence, the expansion of quicklime is suitable for crack control [23,25,29,30,31,32]. Of course, there are other effective shrinkage-reducing admixtures that can cause the expansion. One of them is MgO [33]. However, in some studies of concrete-filled steel tubes, the CaO effect of expansion is more than that of MgO under constant temperature [34]. A similar effect of CaO and MgO admixtures on autogenous shrinkage was obtained in study [28]. However, MgO is more beneficial for massive concrete structures, as it can compensate the thermal-stress-induced shrinkage [29]. According to these results, it was decided to use the CaO admixtures as they are more effective. There are numerous studies evaluating the effect of quicklime [22,23,24,35,36,37,38,39,40,41,42]. However, the usage of quicklime should be limited due to the negative influence on the strength of concrete. In this research, the obtained compressive strength of concrete with 5% quicklime was almost 10% less than that of the same concrete without additives. However, in some of studies, the combination of quicklime and SRA had a greater effect on the shrinkage reduction and less on the compressive strength [11,43]. One combination of quicklime and organic-compound-based liquid SRA was used in this study as well as SRA only. It is known that the quicklime could have an effect on the soundness of cement. Due to this, the soundness of the cement with quicklime additive was tested additionally, and it was found to satisfy the requirements. 

## 2. Materials and Methods

The employed binding material for the concrete mixtures was the CEM I 42.5 R cement, with a fineness of 390 m^2^/kg. Three admixtures were included:Quicklime CL 90 powder, of 300 m^2^/kg fineness and reactivity class R5, complies with the LTS EN 459 standard.Polypropylene microfibers microfiber complies with LST EN 14889-2:2006. The length of fibers varies between 12 and 19 mm, the diameter is 20–35 μm, the density is 0.91 g/cm^3^, and tensile strength varies from 300 to 450 MPa.SRA (Liquid-type shrinkage-reducing agent).

All the concrete mixtures were mixed with the same water/cement ratio (0.48). Moreover, coarse and fine aggregates were included to produce the mixture. Gravel fr. 4/16 mm (of round washed particles, with a density of 2600 kg/m^3^) was included as coarse aggregate, whereas fr. 0/4 mm sand (with a density of 2650 kg/m^3^) was employed as fine aggregate. The same amount of aggregates was used for all concrete mixtures. The particle size distribution of the concrete aggregates was measured as described in EN 933-1 [44] and is given in Figure 1. 

Five different types of mixes were prepared. The first “control” mixture was prepared without any admixtures (Concrete mix No. 1); the rest are described as follows:Concrete mix No. 2—with 0.9 kg/m^3^ of polypropylene microfiber (supplier recommendation, 0.6–0.9 kg/m^3^).Concrete mix No. 3—with 5 wt% quicklime (CaO) powder CL90 (calculated from the amount of cement).Concrete mix No. 4—with 2 wt% SRA (calculated from the amount of cement).Concrete mix No. 5—with 5% quicklime (CaO) and 2% of SRA.

In all the concrete mixtures with or without any admixtures, the ratio of water and cement was the same (W/C = 0.48). Determined and tested concrete mixes can be seen in Table 1.

## 3. Experimental Procedures

### 3.1. Mixing of Specimens

The concrete mixtures and test specimens were created according to the requirements described in LST EN 12390-1:2021 [45] and LST EN 12390-2:2019 [46]. 

The concrete mixture was blended with “Zyklos” mixing equipment. The mixtures were blended for 3 min, as follows: first, the dry materials (aggregates, cement, and microfiber) were added to the mixing bowl and were mixed for 30 s; then, ½ water content was added and mixed for 30 s; subsequently, there was a 1 min break for the better absorption of water; finally, the rest of the water was added together with the plasticizer (shrinkage-reducing additives) and mixed for 1 min. 

For each concrete composition, three prismatic specimens (75 × 75 × 250 mm) were formed for shrinkage measurements and three cubes (100 × 100 × 100 mm) were formed for compressive strength testing (See Figure 2a).

After the fresh concrete was poured and vibrated, it remained in the molds for 24 h (the top of the mold was not covered: the concrete surface was exposed to the environment). After dismantling the molds, the length of the specimens was measured with a shrinkage measurement device (see Figure 3). During the first month after blending, four measurements were performed to determine shrinkage, and later, once a month. 

In order to imitate the closed environment of concrete, a plastic film was used. After the specimens had been disassembled, the measured prisms were covered with 5 layers of plastic film. The prisms were stored indoors at room temperature (18–23°), and not exposed to direct sunlight. During every shrinkage measurement, the plastic film was removed, and a new plastic film was set after the measurement. Hence, during the measurement, the prisms were exposed to the environment (RH—48–54%, C—18–23°) for 3–10 min until they were covered again. An image of a covered prism with plastic film is given in Figure 2b. Concrete cubes were wrapped in plastic film until the day of the compressive strength test. 

### 3.2. Concrete Shrinkage Measurement

The shrinkage measurement was performed according to the standard LST EN 12390-16:2019 [47]. The aim was to determine the length of the prism at certain periods. The measuring equipment precision was ±0.001 mm. The measurement range varied between 0.000 and 13.050 mm. The shrinkage measurement equipment is shown in Figure 3.

Before initiating the measurements, a metal calibration tool was inserted to set the device to zero. A few spins had to be made to ensure values did not change. Then, the investigated concrete prism was inserted, and a few spins were made. During the measurements, the same position of the prism had to be kept (facing the same side and keeping the ends at the same position). The calibration tool was used before and after prism measurements, checking that the measured value is zero. If it was not zero, the measurement had to be repeated.

The concrete shrinkage strain was calculated by applying Equation (1).
(1)$$\varepsilon_{cs}\left(t,t_{0}\right)=\frac{\Delta l\left(t_{0}\right)-l_{cs}\left(t\right)}{L_{0}}$$
where $\varepsilon_{cs}\left(t,t_{0}\right)$ are the total specimen shrinkage strains at time t; $\Delta l\left(t_{0}\right)$ is the initial length of the specimen at time $t_{0}$; $l_{cs}\left(t\right)$ is the length of the specimen at time $t;\;L_{0}$ is the gauge length.

## 4. Tests Results

### 4.1. Compressive Strength Results

The concrete strength was determined by compressing concrete cubes (100 × 100 × 100 mm). Concrete cubes were wrapped in plastic film to imitate a closed environment until the strength test. A concrete strength test was performed after 28 days. To identify the influence of admixtures on concrete strength, five different types of concrete compositions were tested. Three cubes were tested for each composition. The test results are given in Figure 4.

The compressive strength results revealed that some shrinkage-reducing additive reduced the concrete strength, whereas others strengthened the concrete specimens. The control specimen (without additives) exhibited a strength of 49.72 MPa. The addition of polypropylene microfibers produced a slight increase in compressive strength (0.4% higher than the control specimen). On the other hand, the strengths of the specimens including either quicklime or SRA were reduced by 10.1% and 2.6%, respectively. Finally, the specimen with both quicklime and SRA exhibited a slightly lower compressive strength than the control one did (3.7% lower). In a similar investigation carried out by Statkauskas et al. [11], comparable results were depicted.

### 4.2. Flexural Strength Results

The flexural strength of the different concretes was determined by bending shrinkage specimens (75 × 75 × 250 mm). Concrete specimens were wrapped in plastic film to imitate a closed environment for (923 days) and then the flexural strength test was performed. The results in Figure 5 reveal that the control specimen resulted as the weakest one, with 6.4 MPa of flexural strength, i.e., 21.7% weaker than that of the strongest specimen, which was with the microfibers (8.18 MPa flexural strength). The specimen with quicklime additive showed 7.25 MPa of flexural strength (11.7% higher than that of the control specimen). Meanwhile, the specimen with SRA showed 7.29 MPa of flexural strength (13.9% higher than that of the control specimen). The specimen with combined quicklime and SRA revealed a 7.61 MPa flexural strength result (18.8% higher than that of the control specimen). In a similar investigation by Statkauskas et al. [11], comparable results were depicted.

### 4.3. Shrinkage Test Results

The shrinkage of five different concrete compositions was monitored by the method described in Section 3.2. The different types of concrete shrinkage curves are presented in Figure 6. The shrinkage of specimens was monitored for 150 days.

The depicted results revealed that, after 150 days of curing, the greatest shrinkage occurred in the control specimen. Microfibers had a minimum impact on shrinkage reduction, as the specimens only contracted by 1.7% less than the control specimens did. The SRA showed better performance, and the specimen contracted by 17% less than the control one. The concrete mix with quicklime revealed even better results, with the specimen contraction being 50% smaller than that of the control specimen. Finally, the last concrete composition with SRA and quicklime exposed the best contraction reduction results as it showed 54% less contraction compared to the control specimen.

### 4.4. Soundness Test

The Le-Chatelier soundness test was carried out in accordance with the Standard EN 196-3 [48] to evaluate the soundness of cement. Two types of samples were prepared at the normal consistency of cement and cement with 5% of quick lime additive. The paste was prepared, molded, and cured for 24 h in a moist room at a 20° temperature and 90% of relative humidity. After curing, the initial distance between indicator points was measured. Then, the specimens were placed in the water bath and heated gradually to boiling. The specimens in the boiling water were kept for 3 h. Then, the distance between indicator points was measured again. The expansion of both specimens satisfied soundness requirements for cement, i.e., the obtained distance differences were significantly smaller than 10 mm.

## 5. Determination of Concrete Shrinkage

### 5.1. Determination of Initial Data

The prediction of the tested concrete specimen shrinkage can be performed by using some design methods such as EC2 [49] or the model B4 [50,51,52]. For both methods, the compressive cylinder strength is used as the main parameter influencing the concrete shrinkage. Its calculation is further described. First, the mean strength of standard 150 mm sided cubic specimens (*f*_cm,cube_) was calculated by using the reduction coefficient if 0.95 to obtain the required cylinder strength values because, in the tests, smaller non-standard specimens of (100 mm sided concrete cubes) were used. Further, the characteristic cubic strength *f*_ck,cube_ was determined considering a standard deviation of 5 MPa. Finally, the characteristic strength of standard cylinder specimens was calculated by multiplying the characteristic cube strength by the coefficient of 0.8. Hence, the standard strength values resulted as *f*_ck_ = 29.8 MPa and *f*_cm_ = 37.8 MPa. 

### 5.2. Determination of Concrete Shrinkage According to EC2

According to Equations 8–13, B.11, and B.12 given in EC2 [44], the shrinkage of the specimens depends on multiple factors: the compressive strength of concrete (*f*_ck_, *f*_cm_), the type of cement (coefficients α_ds1_ and α_ds2_), the relative humidity (*RH*), and the cross-section parameter *h*_0_. The cross-section of the tested specimens was 75 × 75 mm. The coefficients depending on the type of cement CEM 42.5 N (class N) were α_ds1_ = 4 and α_ds2_ = 0.12. During the test, the first shrinkage measurement was taken immediately after the specimen was removed from the mold. That occurred approximately after 1 day of curing (*t*_0_ = 1 day). The shrinkage strain of the specimens was measured, at least, for 5 months (*t* = 150 days). Taking these data into account, the remaining calculated parameters were the following [49]: $$\beta_{as}\left(t\right)=0.916;\;\varepsilon_{ca}\left(\infty \right)=4.95\cdot 10^{-5};\;\varepsilon_{ca}\left(t\right)=4.53\cdot 10^{-5};\;h_{0}=37.5\;\mathrm{mm};\;k_{h}=1$$

For the shrinkage analysis, the relative external humidity was assumed to be within the range of 70–80%, as the best coincidence between the experimentally determined shrinkage of the control specimen and the shrinkage obtained by the calculation methods occurred when the humidity values were within this range (see Figure 7). 

### 5.3. Determination of Concrete Shrinkage According to Model B4

In a similar way as the EC2 method, the B4 model evaluates the drying and autogenous shrinkage of concrete. However, the B4 model additionally evaluates the composition of the concrete mixture as a significant factor determining the concrete shrinkage. According to Table 1, the parameters related to the concrete mixture are:ac=5.65; wc=0.48; ρ=2357 kg/m3 

*a*—the content of aggregates, 1866 kg; *c*—the content of cement, 330 kg; *w*—content of water, 159 kg.

The parameters related to the “Normal” cement type are [51]:$$\tau_{cem}=0.016;\;p_{\tau a}=-0.33;\;p_{\tau w}=-0.06;\;p_{\tau c}=-0.10;\;\epsilon_{cem}=360\cdot 10^{-6};\;p_{\epsilon a}=-0.80;\;p_{\epsilon w}=1.10;\;p_{\epsilon c}=0.11.\;$$

The autogenous shrinkage parameters related to the “Normal” cement type are [51]:$$\tau_{au,cem}=1.0;\;r_{\tau w}=3.0;\;r_{t}=-4.5;\;r_{\alpha }=1.0;\;\epsilon_{au,cem}=210\cdot 10^{-6};\;r_{\epsilon a}=-0.75;\;r_{\epsilon w}=-3.5.$$

The geometry of the specimen is evaluated by the parameter $k_{s}=1.25$; the parameter depending on aggregates is $k_{\tau a}=1.0$; the specimen-volume–surface ratio is VS=16.3 mm. Further, the constant parameters are calculated as follows: (2)$$E_{28}=4734\sqrt{f_{cm}}=29.11\;\mathrm{GPa};$$
(3)τ0=τcemac6pτawc0.38pτw6.5cρpτc=0.0162 days;
(4)τsh=τ0kτa2ksVS1mm2=26.99 days;
(5)ϵ0=ϵcemac6pϵawc0.38pϵw6.5cρpϵc=4.834·10−4;
(6)ϵau∞=−ϵau,cemac6rϵawc0.38rϵw=−9.70·10−5;
(7)τau=τau,cemwc0.38rτw=2.016;
(8)α=rαwc0.38=1.263.

The abovementioned parameters do not depend on time. The parameters depending on time are presented below, taking the same values of *t*_0_, *t*, and *RH* as in the calculations according to EC2. The two concrete shrinkage curves for relative humidity values 80% and 70% are presented in Figure 8. 

In Figure 8, it can be noticed that the experimental concrete shrinkage curve fits between the two theoretical curves corresponding to 70% and 80% humidity. Considering the results given in Figure 7 and Figure 8, the experimental shrinkage curve is more coincident with the EC2 curve at 70% humidity (see Figure 7), and with the B4 model at 80% humidity (see Figure 8). Immediately after their removal from the mold, the specimens were covered by a polyethylene film to avoid the effect of non-controlled fluctuations in external humidity. During the test in laboratory conditions, the humidity varied from 40% to 55%. While performing each shrinkage measurement, the film was removed for 3–10 min, and then a new film was set. Thus, at each shrinkage measurement time, some moisture of the specimens was lost. Naturally, the highest content of moisture was lost on the first day of curing when the surface of the specimen was not covered by film. The observed curves in Figure 7 and Figure 8 reveal that the calculated shrinkage values by applying both models match accurately with the experimental ones when the relative humidity is within the range of 70–80% of outside relative humidity. However, the real relative humidity of the laboratory was much lower (40–55%). The use of film allowed a constant humidity in the specimens, and that is beneficial as the prediction of shrinkage increment would be more complicated if the humidity changed in time. 

### 5.4. Modification of Model B4 to Predict the Effect of Varying Humidity over Time

One of the ways to predict the shrinkage with changing humidity consists in using an iterative method, whereby the time is divided into separate intervals with separate constant humidity at each interval. The number of intervals can be selected according to the change in humidity in time. The real shrinkage increment will be obtained only for the first interval, whereas for the others, a higher increment will be obtained as the initial time *t*_0i_ does not coincide with the real one *t*_0_. It can be noticed from the shrinkage analysis according to the B4 model that the increments in each time interval are slightly dependent on time, and that these increments are smaller at the beginning of the concrete drying. The same tendency is observed in both methods (see Figure 9 and Figure 10).

However, when these increments are added together, it can be noticed that the shrinkage increment in each interval is similar, with all of them forming a clear straight line. The increments for the tested specimen with 70% outside humidity and 5 days’ time interval are shown in Figure 11. The general line of the shrinkage increments during the whole test time is given in Figure 12. There, for comparison purposes, the real shrinkage curve is presented. The graph shows that the total increment in the intervals is very far from the real one. However, taking into account the linear distribution of the shrinkage increment at each interval, some correction coefficient can be applied to differently reduce the interval increments depending on time. Indeed, this coefficient must evaluate the time from the start of concrete drying. For the calculation of the total shrinkage with the B4 model, such a multiplier $\gamma_{t}\left(t_{i},t_{0i},t_{0}\right)$ can be applied separately in each interval as follows:
(9)$$\epsilon_{sh}\left(t_{i},t_{0i}\right)=\left[\epsilon_{sh\infty }\left(t_{0i}\right)k_{h}\left(t_{i}\right)S\left(t_{i},t_{0}\right)+\epsilon_{au}\left(t_{i},t_{0i}\right)\right]\gamma_{t}\left(t_{i},t_{0i},t_{0}\right),$$
where: (10)$$\epsilon_{au}\left(t_{i},t_{0i}\right)=\epsilon_{au\infty }{\left[1+{\left(\frac{\tau_{au}}{t_{i}}\right)}^{\alpha }\right]}^{r_{t}},$$
(11)$$k_{h}\left(t_{i}\right)=1-RH{\left(t_{i}\right)}^{3},$$
(12)$$S\left(t_{i},t_{0}\right)=tanh\sqrt{\frac{t_{i}-t_{0}}{\tau_{sh}}},$$
(13)$$\gamma_{t}\left(t_{i},t_{0i},t_{0}\right)={\left[\frac{{\left(t_{i}-t_{i0}^{a_{1}}\right)}^{0.965}}{t_{i}-t_{0}}\right]}^{1.66},$$
(14)$$a_{1}=1+\frac{t_{01}^{0.24}}{8000},\;$$

*t*_0*i*_ and *t_i_
*—initial and final interval “*i*” day, respectively; *t*_0_—time when the specimens begin drying.

The final shrinkage is obtained by summing the strains of all time intervals with multiplier $\gamma_{t}\left(t_{i},t_{0i},t_{0}\right)$, according to Equation (15):(15)$$\epsilon_{sh}\left(t,t_{0}\right)=\overset{n}{\underset{i=1}{\sum }}\left[\epsilon_{sh\infty }\left(t_{0i}\right)k_{h}\left(t_{i}\right)S\left(t_{i},t_{0}\right)+\epsilon_{au}\left(t_{i},t_{0i}\right)\right]\gamma_{t}\left(t_{i},t_{0i},t_{0}\right)$$

The obtained concrete shrinkage curves with 70% and 80% humidity by using Equation (15) are given in Figure 13. The model that applies the multiplier $\gamma_{t}\left(t_{i},t_{0i},t_{0}\right)$ is referred to as “the modified B4 model” and is denoted as “B4-mod”.

Figure 13 shows that when the curing time is *t* < 70–80 days, the curves for models B4 and B4-mod do not coincide well. However, such a difference exists only at the initial period of drying. At the time of *t* > 80 days, the coincidence becomes significantly better. This behavior can be noticed more clearly in Figure 14, in which the dependences are given for a longer period of time than in Figure 13.

Figure 14 shows that the difference between the shrinkages calculated according to the B4 model and the B4-mod model becomes minor at later stages of concrete drying. Therefore, it is likely that the shrinkage of the concrete specimen calculated according to B4-mod must also be reasonably accurate for the case of variable humidity. Certainly, this accuracy will depend on the precision with which the variation in humidity is described. Taking into account the fact that the highest content of humidity in concrete must have been lost at the first day of curing without the cover of plastic film, the variation in humidity should be taken as non-linear and decreasing in time. In this research, the humidity is described by Equation (16):
(16)$$RH\left(t_{i}\right)=RH\left(t_{0}\right)-\left(RH\left(t_{0}\right)-RH\left(t\right)\right){\left(\frac{t_{i}}{t}\right)}^{a_{2}},$$
where:(17)$$a_{2}={\left(\frac{1}{t_{i}}\right)}^{0.3},$$
(18)$$RH\left(t_{i}\right)\leq RH\left(t_{0}\right).$$

While analyzing the shrinkage of the tested specimen, the humidity variation was taken from 80% to 70%, and it is shown by the curve of Figure 15.

Finally, the shrinkage curve of the concrete, considering the function of humidity, calculated by the modified B4-mod model was compared with the curves obtained by the B4 and EC2 methods at constant humidity, and also with the curve of the experimental results (see Figure 16). In this figure, it can be seen that, until 40 days of drying, the experimental curve was best coincident with that obtained by EC2 at 70% humidity. However, at later curing stages, the experimental curve starts fitting better with the curve obtained by the B4mod method, when humidity varies non-linearly from 80% to 70%. Naturally, as mentioned before, the time–humidity dependence has a considerable influence on the final curve of shrinkage.

### 5.5. Determination of Shrinkage with Quicklime Additive

The influence of quicklime additive can be evaluated in a similar way as the autogenous shrinkage in EC2 method. According to this, it was assumed that quicklime additive has a limited time effect in the earlier period of drying. Shrinkage investigations of specimens with a high content of quicklime [1,40,41] revealed that the variation in humidity does not produce such a significant residual effect as those of plain concrete specimens. In some of this research [40], it seems that drying shrinkage of specimens with 3–12% quicklime additive develops faster during the first 30 days, and then becomes slower at later curing stages. In fact, after 60 days of curing, the increase in the difference between shrinkage curves of specimens with 3–5% additive quicklime seems to disappear, i.e., the difference becomes constant. Hence, in this research, it is assumed that the effect of quicklime additive on shrinkage takes place during the initial curing period. Therefore, the reduced shrinkage strain due to quicklime additive is calculated through Equation (19):(19)$$\epsilon_{sh}\left(t,t_{0}\right)=\epsilon_{sh\infty }\left(t_{0}\right)k_{h}\left(t\right)S\left(t,t_{0}\right)+\epsilon_{au}\left(t,t_{0}\right)-\epsilon_{lime}\left(t,t_{0}\right),$$
where:(20)$$\epsilon_{lime}\left(t,\;t_{0}\right)=C_{l}{\left(\frac{t}{100}\right)}^{a_{3}},$$
(21)Cl=cL100.25000,
(22)$$a_{3}={\left(\frac{1}{t}\right)}^{0.9},$$

*c_L_*—content of quicklime additive in %.

After the calculation of shrinkage strains according to the B4 model, the theoretical and the experimental curves were compared (see Figure 16). In Figure 16, it can be seen that the experimental shrinkage curve of the concrete specimen with quicklime additive was between the two theoretically obtained by the B4 model with 70% and 80% humidity (B4_70 and B4_80, respectively) as for the control concrete specimen. In this case, the modified B4-mod model allows a theoretical modeling of the shrinkage if humidity decreases in time. The obtained curve with decreasing humidity (B4-mod_70_80_lime) is most coincident with the experimental one (see Figure 17).

## 6. Discussion

The application of the multiplier $\gamma_{t}\left(t_{i},t_{0i},t_{0}\right)$ is a convenient and simple way to evaluate the influence of humidity reduction over time on concrete shrinkage. In the current study, the application of this multiplier was verified for a relatively narrow range of humidity corresponding to the case of partially closed-section steel-concrete structures. However, it may be the case that for wider humidity ranges, the multiplier should be slightly corrected. In any case, the modified B4 model (B4-mod) was checked with cycling humidity changes. As mentioned in the introduction, some studies have found that the shrinkage of concrete with cyclic changes in humidity is similar to the shrinkage obtained at a constant average humidity. Under the conditions described in the study (when humidity decreases from 80% to 70%), the shrinkage after 250 days of drying for the case of cyclic humidity is 0.000403, a similar value to that obtained in the case of constant 75% average humidity: 0.000392. Hence, the difference is less than 3%. In the B4-mod model, the cyclic humidity was described as a time-dependent sinusoidal function, varying in a humidity range between 70% and 80%. If the humidity decreases from 80% to 40%, the concrete shrinkage with cyclic humidity is 0.000512, while for a constant 60% average humidity, it is 0.000496 (differing only by 3%). From this point of view, there is no significant shrinkage difference between the narrow and the wide range of humidity reduction. Another tendency of the model is noticed when the humidity over time decreases continuously, and not cyclically. In the case of the linear decrease of humidity, the concrete shrinkage is 0.000353, when the humidity decreases from 80% to 70%, and 0.00037, when the humidity decreases from 80% to 40%. In these cases, the difference between the obtained shrinkage values and the shrinkage at average constant humidity (75% and 60%) increases to 11% and to 34%, respectively. Comparing these differences with the ones under cyclic humidity changes, it becomes clear that the kind and range of humidity change influence the shrinkage according to the B4-mod model. Without additional experiments, it would be difficult to answer the question of whether the humidity at the beginning of drying is much more important than at the end. Moreover, a noticeable difference in concrete shrinkage is obtained between the linear and parabolic humidity reduction from 80% to 40%. In this case, the shrinkage with the linear humidity change is 0.00037 and, with the parabolic one (as presented in Figure 15)—0.000417. The difference is 13%, considerably higher than the 4% obtained when the humidity decreases from 80% to 70%. In order to verify such concrete shrinkage tendencies, some experimental studies must be carried out with sufficiently precise control of variable humidity. Nevertheless, the B4-mod model is a suitable way to predict the concrete shrinkage, at least in the narrow range of humidity variation. 

## 7. Conclusions

Five different concrete compositions were prepared to evaluate the effect of different admixtures on the shrinkage of concrete. After 150 days of curing, the greatest shrinkage reduction was depicted for concrete including both SRA and quicklime additives. For this concrete, the shrinkage decreased by 54.0%, in respect of the control specimen. The concrete mix with 5% of quicklime reduced the shrinkage up to 50.0%, and with 2% of SRA—up to 17,0%. In comparison, the effect of the polypropylene fibers was negligible (reduction of 1.7%). 

The differences between the compressive strength values of the various concrete specimens were not so significant as those between shrinkage values. In all cases, the difference between the compressive strength of the specimens and the strength of the control one was smaller than the standard deviations (<5%), except in the case of the specimens with quicklime additive, which reduced the compressive strength by 10.1%. Considering that the EC2 method mostly evaluates the compressive strength of concrete, shrinkage values do not show a significant difference. 

The external conditional humidity was set from 70% to 80% after a comparison of the experimental shrinkage measurements and the shrinkage curves calculated by EC2 and B4 models for the control specimen. The theoretical concrete shrinkage results obtained by model B4-mod shows that this method is suitable to accurately predict shrinkage. After 150 days of concrete curing, the differences between the experimental shrinkage results and the obtained ones by the modified B4 model reached up to 3%. 

In addition, the modified B4 model (B4-mod) can be applied to predict the shrinkage of concrete including a certain quicklime additive content. In this study, it was assumed that the effect of quicklime on the shrinkage depends on the concrete composition. Thus, the shrinkage strain component may have to be improved to adapt this component to other compositions of concrete. For the concrete specimen with 5% quicklime additive, the experimental shrinkage curve was best coincident with the theoretical one obtained by the modified B4-mod. After 150 days of concrete with a quicklime additive curing, the differences between the experimental shrinkage results and the obtained ones by B4-mod model reached up to 12%. 

## Figures and Tables

**Figure 1 materials-16-02061-f001:**
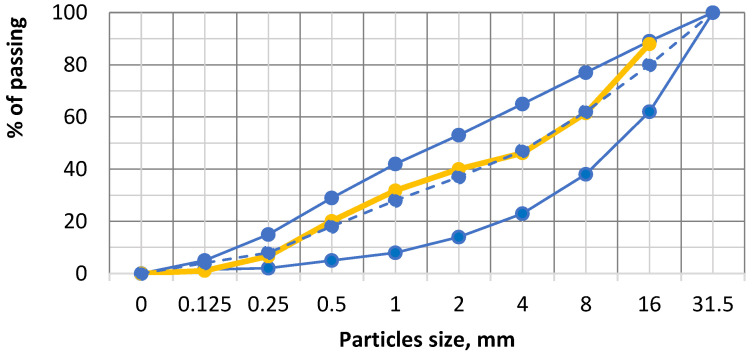
Gravel and sand granulometry. Continuous lines—upper and lower limit for concrete aggregate. Dashed line—recommended concrete aggregate distribution. Yellow line—tested aggregate results.

**Figure 2 materials-16-02061-f002:**
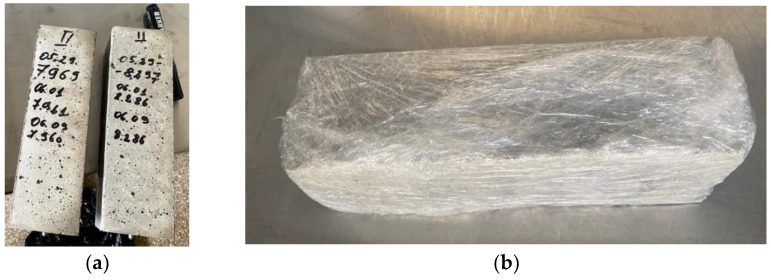
Formed concrete specimens: prisms of 75 mm × 75 mm × 250 mm during shrinkage measurement (**a**); same prism during storage period wrapped in the plastic film (**b**).

**Figure 3 materials-16-02061-f003:**
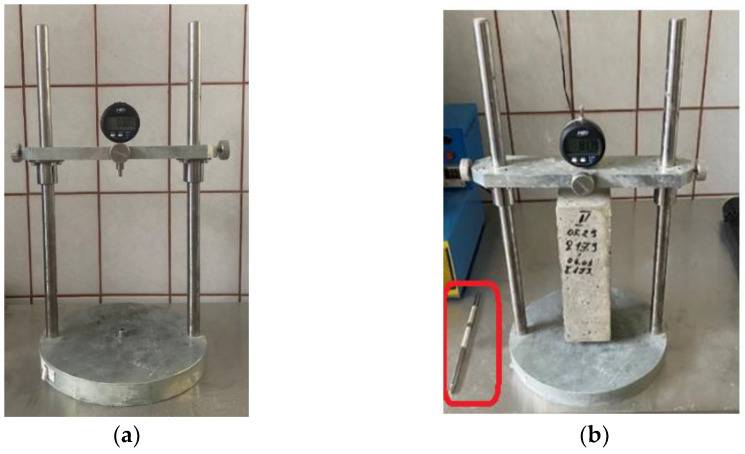
Views of the measurement device of the concrete prism shrinkage: without a specimen (**a**); with a specimen (**b**). Calibration tool is marked in red.

**Figure 4 materials-16-02061-f004:**
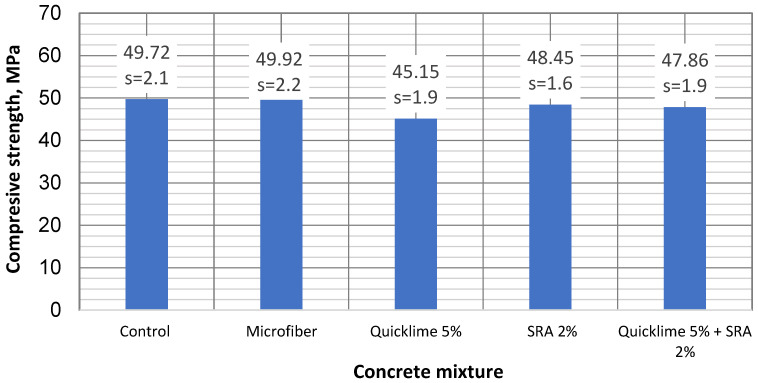
The compressive strength results of various concrete compositions.

**Figure 5 materials-16-02061-f005:**
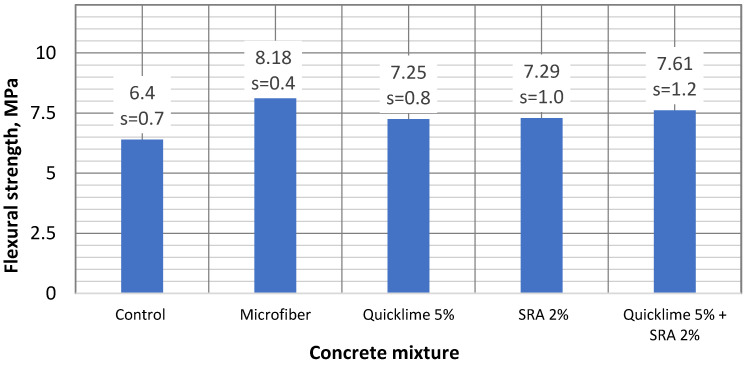
The flexural strength results of various concrete compositions.

**Figure 6 materials-16-02061-f006:**
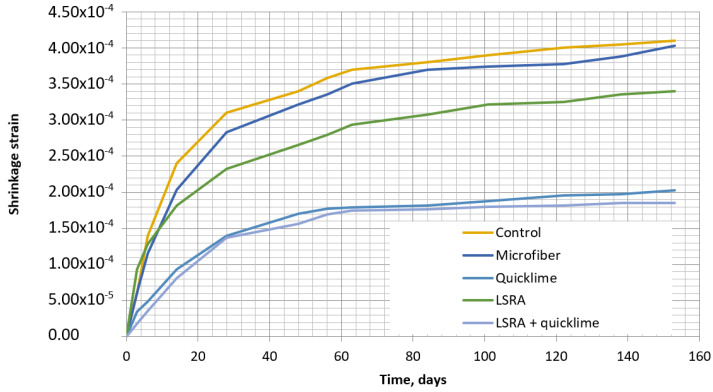
Dependences of shrinkage deformations of the various specimens on curing time.

**Figure 7 materials-16-02061-f007:**
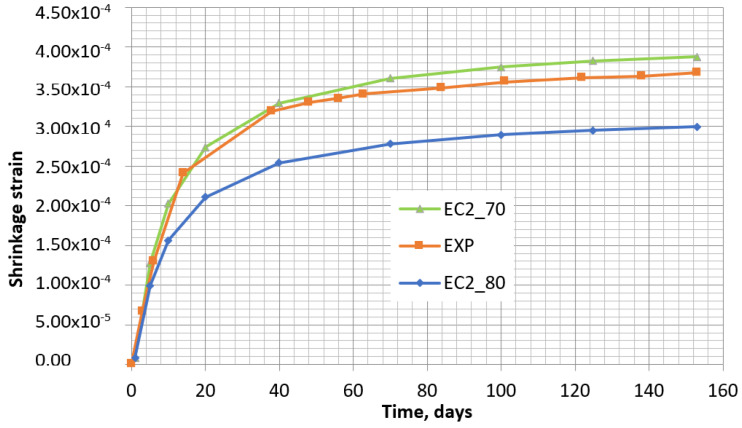
Shrinkage of the control specimen obtained experimentally (EXP) and by EC2 design method, at relative humidity of 70% (EC2_70) and 80% (EC2_80).

**Figure 8 materials-16-02061-f008:**
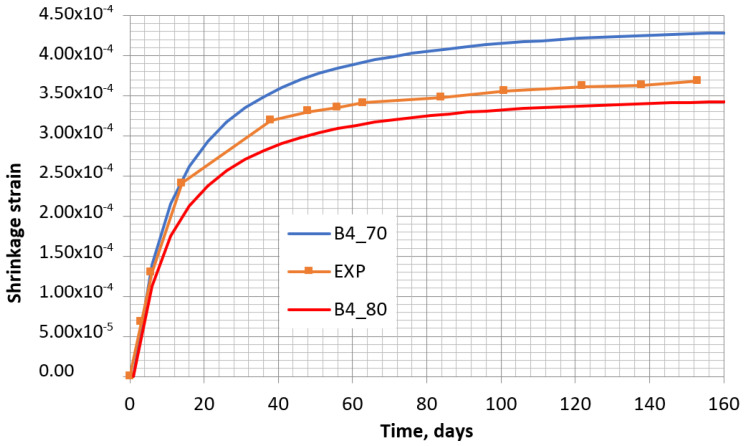
Shrinkage of the control specimen obtained experimentally (EXP) and by B4 model, at humidity of 70% (B4_70) and 80% (B4_80).

**Figure 9 materials-16-02061-f009:**
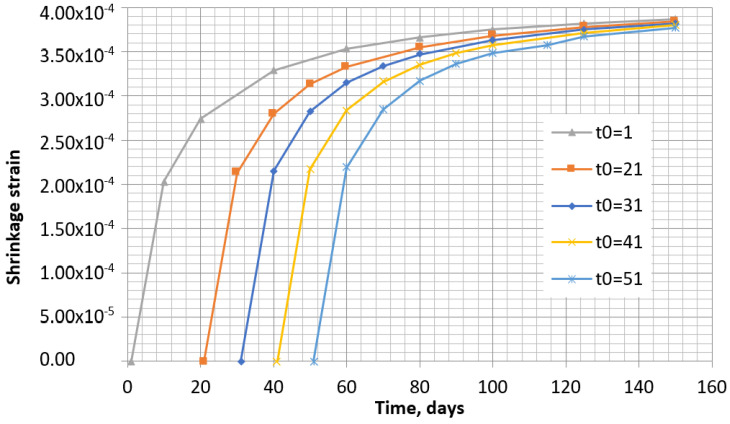
Shrinkage of concrete specimens according to EC2 model at the different drying start times when humidity is 70%.

**Figure 10 materials-16-02061-f010:**
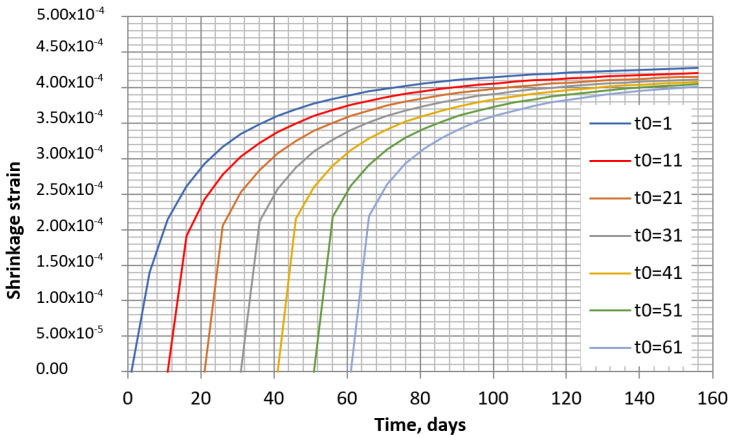
Shrinkage of concrete specimens according to B4 model at the different drying start times when humidity is 70%.

**Figure 11 materials-16-02061-f011:**
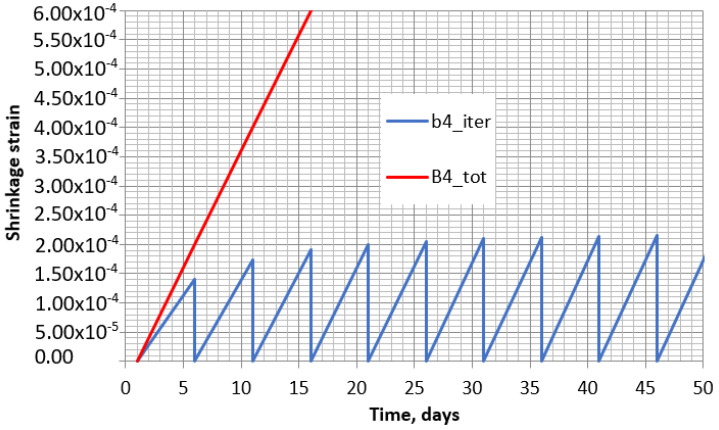
Shrinkage of each separate interval (B4_iter) and their sum (B4_tot) obtained according to model B4 with 70% external humidity.

**Figure 12 materials-16-02061-f012:**
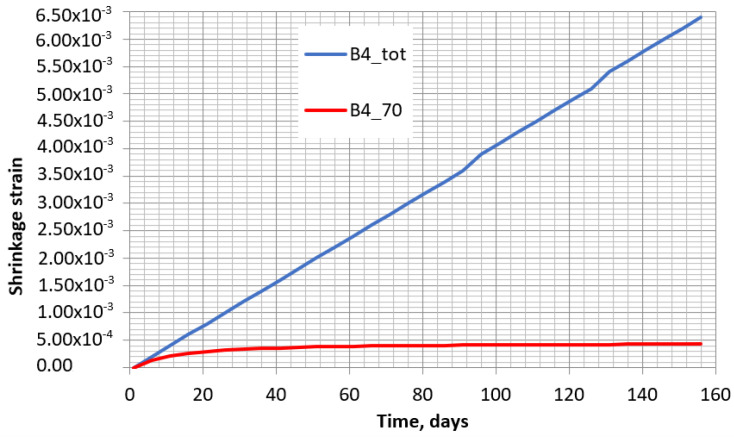
Shrinkage obtained according to B4 model (B4_70) and sum of shrinkage at each interval (B4_tot) with 70% humidity.

**Figure 13 materials-16-02061-f013:**
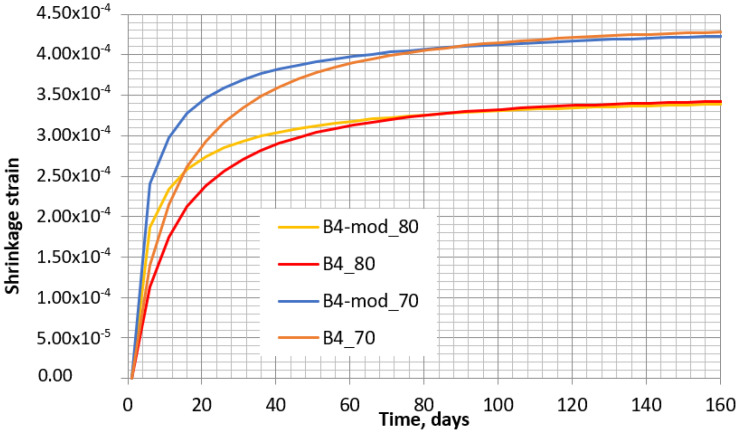
Shrinkage at humidity of 70% and 80% according to B4 model (B4_70 and B4_80, respectively) and modified B4-mod model (B4-mod_70 and B4-mod_80, respectively) in 160 days period.

**Figure 14 materials-16-02061-f014:**
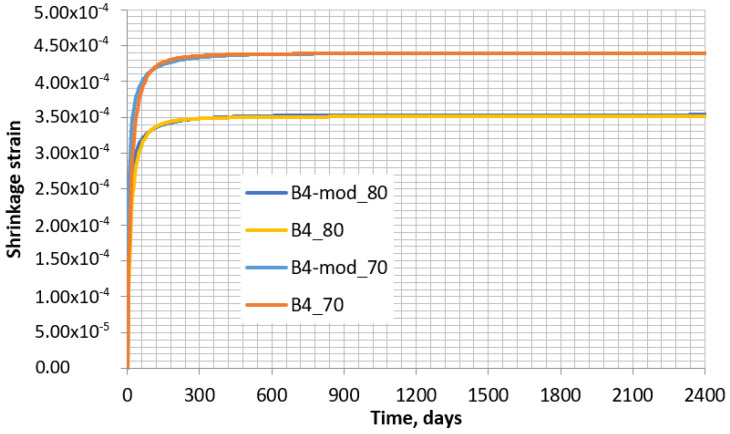
Shrinkage at humidity of 70% and 80% according to B4 model (B4_70 and B4_80, respectively) and modified B4 model (B4-mod_70 and B4-mod_80, respectively) in a 2400-day period.

**Figure 15 materials-16-02061-f015:**
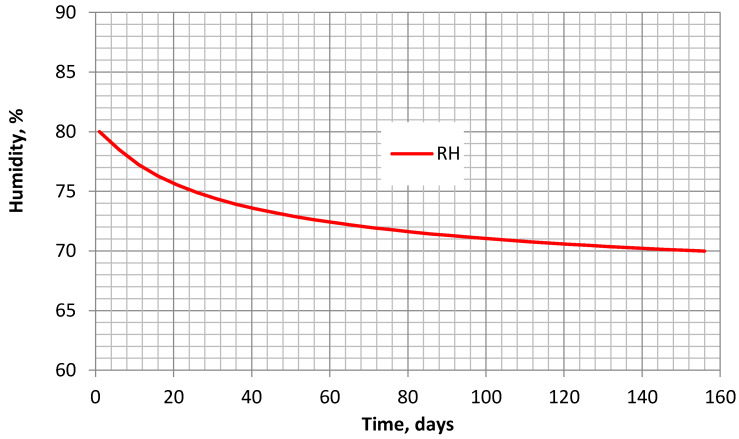
Dependence of humidity (from 80% to 70%) on time.

**Figure 16 materials-16-02061-f016:**
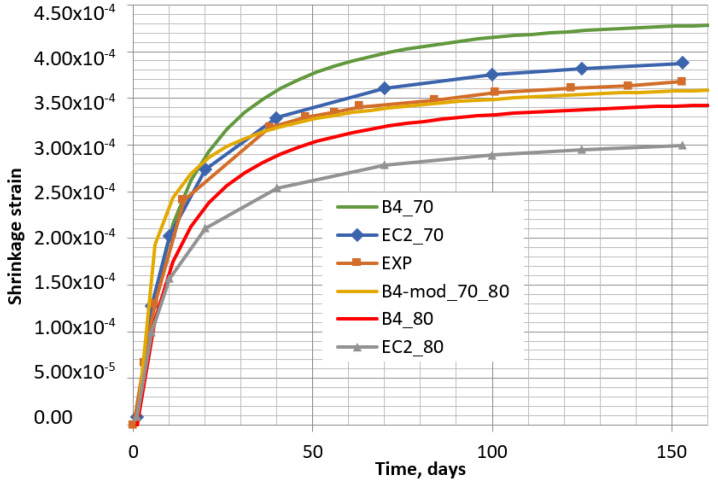
Shrinkage of control specimens obtained experimentally (EXP) and by methods EC2, B4, and B4-mod with 70% and 80% humidity (EC2_70, EC2_80, B4_70, B4_80, and B4-mod_70_80).

**Figure 17 materials-16-02061-f017:**
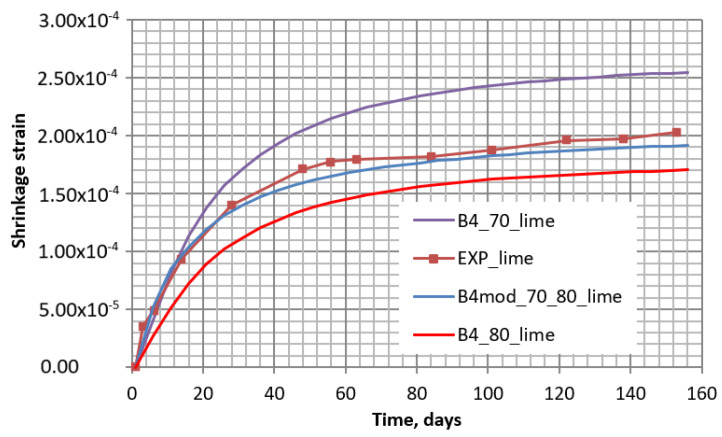
Shrinkage of specimens with quicklime additive obtained experimentally (EXP_lime) and by methods B4 and B4-mod with 70% and 80% humidity (B4_70_lime, B4_80_lime, and B4-mod_70_80_lime, respectively).

**Table 1 materials-16-02061-t001:** Concrete compositions capable of dealing with shrinkage.

	Concrete Mixture	1 (Control)	2 (Microfiber)	3 (Quicklime)	4 (SRA)	5 (Quicklime+ SRA)
Composition						
▪ Cement CEM I 42.5R, kg/m^3^		330				
▪ Gravel 4/16, kg/m^3^		976				
▪ Sand 0/4, kg/m^3^		890				
▪ SRA, 0.55% (kg)		1.82				
▪ Water, kg/m^3^		159				
▪ W/C		0.48				
▪ Polypropylene microfiber, kg		-	0.9	-	-	-
▪ Quicklime, 5%, for cement quantity, kg/m^3^		-	-	16.5	-	16.5
▪ SRA, 2% for cement quantity, kg/m^3^		-	-	-	6.6	6.6

## Data Availability

The data presented in this study are available on request from the corresponding author.

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
