# Peer review of "Concrete Shrinkage Analysis with Quicklime, Microfibers, and SRA Admixtures"

_materials, 2023, doi:10.3390/ma16052061_

Round 1

Reviewer 1 Report (New Reviewer)

In this research article, authors have worked on "Concrete shrinkage analysis with quicklime, microfibers and RSA admixtures". Suggested to incorporate the following comments in the revised submission.

1. The research carried out in this paper is quite good. However, the way the article is presented by the authors may not be very clear to the intended audience. In addition, many formatting errors (alignment, bolds, caps, subscripts, and superscripts) are also present in the paper. The reviewer recommends formatting and revising the paper properly and resubmitting it with the corrections mentioned in the subsequent paragraphs.

2. The introduction provides a good background of the research. However, it should be written better way to provide a clearer and more crisp background about this specific topic to date so that it will be easily understandable to the audience of this paper.

3. On what basis authors have decided on the dosage of admixture, lime, and fiber?

4. Check Figure 1, Y-axis. You can write in simple English like "% of passing"

5. Check Figure 2, X-axis. The prism size will be 75x75x500mm, not 275mm.

6. Compare the results and discussion with available literature. You can refer to the following manuscripts that worked on shrinkage, and creep with various admixtures and incorporate them as references in the revised manuscript.

A) Shruthi, V. A., Shwetha, K. G., Nagendra, R., Ranganath, C., Ganesh, B., & Mahesh Kumar, C. L. (2021). Strength and drying shrinkage of high strength self-consolidating concrete. In Recent trends in civil engineering (pp. 615-624). Springer, Singapore.

B) Tangadagi, R. B., Ganesh, B., Vasudev, M. V., Nagendra, R., & Ranganath, C. (2021). Creep characteristics of high strength self compacting concrete. In Recent Trends in Civil Engineering (pp. 625-635). Springer, Singapore.

C) Tangadagi, R. B., Seth, D., & Preethi, S. (2021). Role of mineral admixtures on strength and durability of high strength self compacting concrete: An experimental study. Materialia, 18, 101144.

7. The conclusion section should be properly rewritten with both qualitative and quantitative explanations.

8. Incorporate at least 5-6 references from MATERIALS journal.

9. Remarks

The study carried out by the authors in this paper has novelty and contributes significantly to the current state of research. But the manuscript must also be presented in a better way to avoid ambiguity and lack of clarification among the intended audience. The reviewer, therefore, suggests revising and resubmitting the present manuscript for further consideration. 

Author Response

Answer to reviewer comments was provided at word document.

Reviewer 2 Report (Previous Reviewer 4)

This work deal with the shrinkage properties of concrete improved by the admixtures under different humidity, which is interesting and significant. There are some problems should be addressed before its acceptance.

1.     What does “RSA” refer to? In addition, there are some abbreviations in the abstract without full name.

2.     In the introduction, the authors are recommended to introduce experimental works first and then predictive model. In addition, more references about the SRA should be added, such as Influences of MgO and PVA fiber on the abrasion resistance, cracking risk, pore structure and fractal features of hydraulic concrete, Application of shrinkage reducing admixture in concrete: A review, Properties of high-performance concrete containing shrinkage-reducing admixture, etc.

3.     The title of vertical axis of figure 1 should be changed.

4.     At line 128-129, the unit of the size of specimen should be given. In other part of manuscript, some specimen size are not specified with unit. Please check.

5.     The shrinkage measurement is kept for 310 mins. Is this duration according to any standards or specified by the authors?

6.     In figure 4, please represent standard deviation with another symbol since σ usually represents stress.

7.     Ιn figure 5, dot should be used instead of comma.

Author Response

Answer to reviewer comments was provided at word document

Reviewer 3 Report (Previous Reviewer 3)

The manuscript was rejected in a previous version.

The new version is redesigned and has good quality.

The manuscript is well structured.

Possible improvements to the manuscript include that the discussion of results and conclusions can be improved.

For further editing, focus on the visual of the manuscript. This is also very important.

Author Response

Thank you for the notes, slight adjustments were made.

Round 2

Reviewer 2 Report (Previous Reviewer 4)

The introduction part should be well refined. This paper is concerning about concrete shrinkage  with quicklime, microfibers and  SRA admixtures, hence, more related experimental studies of effects of fiber, MgO, SCMS such as fly ash and other SARs, for instance, Effects of fly ash dosage on shrinkage, crack resistance and fractal characteristics of face slab concrete; Influence of MgO on the hydration and shrinkage behavior of low heat Portland cement-based materials via pore structural and fractal analysis,  should be introduced, so as the readers could understand better why the authors used quicklime and microfibers here. The current introduction part is insufficient. More related and new studies should be covered.

Besides, the authors claimed that “There are various types of SRA, such as fibers, microfibers, ab- 55 sorbent polymers and supplementary cementitious materials.” In my opinion, the SRA refers to some chemical mixtures, rather than fiber and SCM. Please recheck the notion of SRA.

Author Response

Thank you for your insights, the manuscript was updated accordingly, and more in-depth answers are provided in the attached word file.

This manuscript is a resubmission of an earlier submission. The following is a list of the peer review reports and author responses from that submission.

Round 1

Reviewer 1 Report

In this manuscript, the authors explored the effect of various humidity conditions and efficiency of shrinkage reducing additives to the shrinkage strain of ordinary Portland cement concrete and its mechanical properties. Ordinary C30/37 concrete mixture was remanufactured with micro-polypropylene fibers or quicklime and/or sike control 50. The manuscript seems to be within the scope of the journal. However, the manuscript revealed lack of relevant scientific novelty and the results contribute very little to the state of the art. The experiment is relatively basic and simple. At the same time, this work does not deeply analyze the mechanism of the difference in mechanical properties. The number of specimens conducted per test were three specimens/mix, not enough to draw a conclusion or state a scientific discussion. Overall, the manuscript is written in an average English and it needs a thorough review. The figures are presented in report like style. Hence, I do not recommend the manuscript to be published in its current form.

The following some specific comments need to be taken into account to further improve the quality of the manuscript:

Comment 1: The paper should be carefully double-checked from grammatical point of view. 

Comment 2: What is the main objective behind the current study? It is beneficial for the readers to add more explanations about the novel contribution of this method from theoretical/experimental viewpoints.

Comment 3: Figures 1 (a) that show aggregates view are not necessary.

Comment 4: Table 1: Correct V/C to W/C

Comment 5: page 3-lines (88-113): the description of the different concrete mixtures is redundant and can be better summaries and presented.

Comment 6: page 4-section 3.1: Please clarify at which step the cement was introduced to the mixture.

Comment 7: page 5-Figure 5: Please improve the quality/presentation of the figure. Moreover, error bars (standard deviations) should be provided.

Comment 8: page 6-Figure 5: the variations on the compressive strength between control mix, microfiber, and quick lime + sika control mixes are not significant and within experimental error (0-2%), hence, conclusion cannot be drawn here.

Comment 9: page 6- lines (188-195): the author compared the shrinkage results of the different concrete mixtures by providing some percentages. But, it is not clear at which stage/age this comparison was considered.

Comment 10: page 7- lines (199) correct reference.

Comment 11: page 7- lines (218): what are other parameters represent?

.

.

Comment 12: The conclusion part needs to be refined and more precise.

Author Response

Comments are provided in the word file.

Reviewer 2 Report

At the end of the abstract, the experimental shrinkage curve best coincident with theoretical one by modified B4 model can be added.

In figure 5, the title in the figure should match the title given.

Line 269, “From the shrinkage analysis according to B4 model can be seen that the increments in each time interval a little bit depending on time and these increments are less at the beginning of concrete drying.” I think that these increments are bigger at the beginning of concrete drying. You could have a check.

The language is too weak, three are too many mistakes in this paper. I can not even find out one sentence without a grammar mistake. It is strongly suggested to refine the language under the help of a native speaker or language editing service. Unless this paper is revised word by word, I will not propose acceptance.

Polypropylene microfiber and Quicklime are used in this paper. Why did the authors select such materials? Why not some other common fibers? More specific reasons and statements should be given in the introduction. Are there any similar studies in the world.

It is confusing to use lime in concrete, since it can easily cause shrinkage and cracking. Are there any similar studies in the world? Thus, the introduction should be supplemented with many new and relative references to support this paper.

Some shrinkage and cracking-prevention related papers such as Effects of Fly Ash Dosage on Shrinkage, Crack Resistance and Fractal Characteristics of Face Slab Concrete; Influence of reactivity and dosage of MgO expansive agent on shrinkage and crack resistance of face slab concrete; The influence of fiber type and length on the cracking resistance, durability and pore structure of face slab concrete;

Please give more details about the experiments in this study. These papers used fly ash, fibers, and MgO rather than lime to reduce shrinkage.

It is improper to place the pictures of some common specimens in this paper, such as Figures 2 and 3. 

In figure 5, the strength difference among different mixtures are so minor to distinguish different effects of mineral materials. How about the errors.

Please explain the reason why should conduct the “5. Determination of concrete shrinkage”, what is the purpose. Can you tell the readers the reason in the Introduction part?

Author Response

Comments are provided in the word file.

Reviewer 3 Report

Shrinkage is a very topical research topic. 

The research area belongs to the limits of a very important research area, where the research must be solved in the context of a complex experimental program.

The manuscript has the usual structure, but some parts are missing. It is the discussion part. 

The presentation of new knowledge must be significantly improved in the manuscript. In the introduction part, there is very little information from the addressed area. 

Only 26 links are listed. Must be improve. 

for example from MDPI:

Microstructure, Shrinkage, and Mechanical Properties of Concrete with Fibers and Experiments of Reinforced Concrete Beams without Shear Reinforcement

;

Preparation and Experimental Investigations of Low-Shrinkage Commercial Concrete for Tunnel Annular Secondary Lining Engineering

many others.

State the motivation for the addressed area more clearly. 

There are mistakes and typos in the manuscript. 

Check the format of the references according to the MDPI template. 

There is an invalid link on page 6. 

I would encourage the authors to significantly improve the processing of graphs. 

It must have a uniform method of processing - format of axes, division of graphs. somewhere it is 150, somewhere it is 160. Fig. 15 and 16, and others. 

In the manuscript, make a part - a list of abbreviatio.

The experiments and tests performed are interesting in manusript. 

However, it would be advisable to expand the experimental program (recommendations only). An ideal experimental program should include compressive strength, tensile strength, modulus of elasticity, fracture mechanical parameters or microstructural details. 

I recommend that authors carefully revise and rewrite the manuscript. The topic addressed is very interesting and the results in the manuscript will potentially be interesting for readers as well.

Author Response

Comments are provided in the word file.

Reviewer 4 Report

This paper tents to study the shrinkage of concrete with admixtures. This paper should be improved too much before it can be accepted.

1) There are too many admixtures. The title should mention the specific materials this paper used, that is, the Polypropylene microfiber, lime and Sika Con-9 trol should be given.

2) What is the novelty of this study. The author should keep in mind what are the differences between this paper and others that make this paper new and novel. The authors should make a careful comparison.

3) What is the main purpose of this paper. From the review in the introduction, we can see some researchers have already conducted the similar studies.

4) It is widely known that lime or CaO should be avoided in concrete to prevent the durability and shrinkage problems. How can the authors use this material in concrete? Can they add some similar studies to validate this technique?

5) This paper should be revised by a native whose mother language is English.

6) The authors should enrich the reference part, because as far as I know, the materials used in this paper are not common ones to reduce shrinkage. The authors should follow the widely techniques.

Author Response

Comments are provided in the word file.

Round 2

Reviewer 1 Report

no comment

Author Response

No additional comments were provided.

Reviewer 2 Report

Many of previous comments are not carefully and fully addressed.

Especially the introduction part should be improved.

Author Response

The answer is provided in the word document.

Reviewer 3 Report

The manuscript has been improve. 

Selected parts and details of the manuscript are improved. Unfortunately, the overall quality of manuscript does not reach the required level. Authors should revise and improve the manuscript overall:

1) More details about the experimental program must be given.

2) The discussion of the results must be improved.

3) Focus on describing new knowledge. This is related to the originality of the topic.

4) The manuscript should be revised by an English speaker, e.g. at MDPI.

Author Response

(The authors gave the same response as above.)

Reviewer 4 Report

The revision is weak.

Author Response

(The authors gave the same response as above.)
